# Human-Induced Pluripotent Stem Cell Technology: Toward the Future of Personalized Psychiatry

**DOI:** 10.3390/jpm12081340

**Published:** 2022-08-20

**Authors:** Alessandra Alciati, Angelo Reggiani, Daniela Caldirola, Giampaolo Perna

**Affiliations:** 1Department of Clinical Neurosciences, Villa San Benedetto Menni—Hermanas Hospitalarias, Via Roma 16, 22032 Albese con Cassano, Como, Italy; 2Humanitas Clinical and Research Center, Via Manzoni 56, 20089 Rozzano, Milan, Italy; 3D3 Validation Research Line, Istituto Italiano di Tecnologia, Via Morego 30, 16163 Genova, Italy; 4Department of Biological Sciences, Humanitas University, Via Rita Levi Montalcini 4, Pieve Emanuele, 20090 Milan, Italy

**Keywords:** personalized psychiatry, psychiatric disorders, induced pluripotent stem cells, brain organoids

## Abstract

The polygenic and multifactorial nature of many psychiatric disorders has hampered implementation of the personalized medicine approach in clinical practice. However, induced pluripotent stem cell (iPSC) technology has emerged as an innovative tool for patient-specific disease modeling to expand the pathophysiology knowledge and treatment perspectives in the last decade. Current technologies enable adult human somatic cell reprogramming into iPSCs to generate neural cells and direct neural cell conversion to model organisms that exhibit phenotypes close to human diseases, thereby effectively representing relevant aspects of neuropsychiatric disorders. In this regard, iPSCs reflect patient pathophysiology and pharmacological responsiveness, particularly when cultured under conditions that emulate spatial tissue organization in brain organoids. Recently, the application of iPSCs has been frequently associated with gene editing that targets the disease-causing gene to deepen the illness pathophysiology and to conduct drug screening. Moreover, gene editing has provided a unique opportunity to repair the putative causative genetic lesions in patient-derived cells. Here, we review the use of iPSC technology to model and potentially treat neuropsychiatric disorders by illustrating the key studies on a series of mental disorders, including schizophrenia, major depressive disorder, bipolar disorder, and autism spectrum disorder. Future perspectives will involve the development of organ-on-a-chip platforms that control the microenvironmental conditions so as to reflect individual pathophysiological by adjusting physiochemical parameters according to personal health data. This strategy could open new ways by which to build a disease model that considers individual variability and tailors personalized treatments.

## 1. The Limitations of Personalized Medicine in Psychiatry

Personalized medicine aims to predict disease susceptibility, achieve an accurate diagnosis, and select the most effective therapeutic option with the least adverse effects for each patient [1,2]. Personalized predictive models of diagnosis and outcomes are based on the collection of a massive amount of information (big data) to analyze using advanced computational tools capable of making a prediction from the gathered data, such as machine learning methods [2].

Before its psychiatric application, personalized medicine had brought significant diagnostic and therapeutical advances in oncology where the identification of the molecular target of the drug, the mechanisms of resistance to targeted compounds, and the best pharmacological combinations for a specific tumor promoted the optimal individual treatment selection [3].

Due to the lack of practicable diagnostic tests, along with the inherent complexity of psychiatric disorders, the reliance on diagnostic manuals make personalized medicine in psychiatry implementation more challenging than in other medical specialties [4]. The currently available diagnostic tools, the Diagnostic and Statistical Manual of Mental Disorders (DSM-5) [5] and the International Statistical Classification of Diseases and Related Health Problems (ICD-11) [6], provide a classification that fails to capture the biological processes that are believed to underlie the identified disorders and favor treatment personalization in psychiatry [7]. This is partly because even the disorders more narrowly defined with diagnostic tools generally represent heterogeneous endpoints of different underlying causal pathways, that are amply intertwined with social, cultural, and experiential factors [8].

A diagnostic system using a combination of several psychopathological dimensions defined along with a range of severity, from normality to the full pathological expression, has been proposed because the categorical approach fails to fully express mental illness complexity [9]. However, research evidence demonstrating that a patient evaluated using a dimensional approach improves with treatment is lacking. Replacing the categorical DSM-5/ICD-11 classification system with a dimensional one appears to not be feasible if clinical outcomes are not demonstrably better [10].

Animal models have widely contributed to understanding several brain pathological mechanisms and developing effective drugs; however, fully reproducing the complexity of human psychiatric diseases is substantially impossible [11]. Moreover, some biological limitations derive from species brain differences. Relative to the mouse, the dominant model organism in research, the human cortex has a >1000-fold greater number of neurons and contains an inner fiber layer and outer subventricular zone that are absent in the mouse brain [12]. Notably, genes associated with neuronal structure and function have different transcriptional regulations in mice, non-human primates, and humans [13]. Moreover, mouse models are adequate to demonstrate pathological consequence production by the altered single gene function, whereas studies of psychiatric disorders, generally characterized by different gene penetrance and complex interactions, are very difficult to perform using mouse models.

Finally, research has been hindered by the inaccessibility to human live brain tissue and the difficulty in isolating and culturing primary human neurons. Postmortem brain studies are burdened by confounding variables, such as treatment history, drug or alcohol abuse, and cause of death, and cannot provide information on temporal psychiatric disease changes in which molecular and cellular alterations may occur years before the clinical symptom presentation [14].

The discovery of induced pluripotent stem cell (iPSC) technology has opened unprecedented opportunities as the limitations described above underscore the need for psychiatric disease models based on human cells. In the next sections, we summarize the main concepts concerning iPSCs and provide an overview of their preliminary applications in the field of personalized psychiatry (PP).

## 2. The Induced Pluripotent Stem Cells

### 2.1. Definition and Development

These iPSCs consist of artificial stem cells that are induced (or “reprogrammed”) in culture from somatic cells using different transcription factors. Stem cells are pluripotent cells derived from the embryo (embryonic stem cells [ESCs]) and various postnatal tissue sources, characterized by the ability of self-renewal (proliferation) and differentiation into various adult cell types, and thus called pluripotent stem cells.

In 2006, Takahashi and Yamanaka [15] demonstrated for the first time that terminally-differentiated somatic cells could be reverted into undifferentiated pluripotent cells and thereby produced a paradigm shift in developmental biology, breaking the previous view of cellular differentiation as a unidirectional and irreversible process. Somatic cells were reprogrammed to iPSCs through genomic integration by retrovirus of the transcription factors (TFs), Oct4, Sox2, Klf4, and c-Myc. The TFs remodel chromatin to activate gene expression in the pluripotency network and suppress lineage-specific gene expression that promotes differentiation [16]. The iPSCs can form neural progenitor cells (NPCs) that can be further stimulated to differentiate into various central nervous system (CNS) cell types, including astrocytes and functionally active neurons. Human iPSCs were first established from skin fibroblasts through a skin biopsy [17]. More recently, peripheral blood mononuclear cells [18], which can be readily attained via phlebotomy, and even cells in the urine have been reproducibly converted into iPSCs [19].

Retroviruses are extensively used for integrating exogenous genes into the genome to reprogram somatic cells and produce iPSCs. However, some risks could limit their use in iPSC-based therapies, such as those proposed to regenerate cardiac cells after myocardial infarction [20] or insulin-producing β cells to treat type 1 diabetes mellitus [21]. Particularly, the over-expression of exogenous reprogramming factors, such as the oncogene c-Myc, could stimulate cancer growth and cause tumor development after iPSC-derived cell transplantation [22]. Strategies to produce iPSCs with minimal or absent genomic integration use lentiviruses, which show a safer integration profile [23], and Sendai virus, an RNA virus that exclusively replicates its genome in the cytoplasm [24]. A non-viral system, such as a DNA plasmid that carries the reprogramming factors, which usually do not integrate into the host genome, is also available. DNA plasmid limitation includes low transfection efficiency compared to traditional systems based on retroviral or lentiviral vectors due to the transient gene expression. Most DNA-based methodologies for reprogramming somatic cells leave foreign genetic materials in the genome. The mRNA transfection is directly translated into functional proteins and does not lead to any host genome modification.

Over the recent years, iPSC application has been frequently associated with gene editing. This technique targets the disease-causing gene to deepen illness pathophysiology and conduct drug screening [25]. Gene editing is a group of technologies that, differently from traditional transgenic techniques, allow to add, remove or alter genetic material at specific genome locations [26]. An accurate and efficient approach to gene editing is CRISPR-Cas9, the acronym for clustered regularly interspaced short palindromic repeats and CRISPR-associated protein 9. The CRISPR/Cas9 system contains two key molecules: the guide RNA (gRNA), which binds to a specific target DNA sequence, and the Cas9 nuclease, which generates a double-stranded DNA break. The DNA repair machinery can then be hijacked to insert one or more DNA sections into the genome [27]. CRISPR-Cas9 is generally used in iPSCs to generate cell models to investigate CNS disorders with a defined genetic alteration. Moreover, the identified pathogenic mutation in these models can be corrected by inserting, deleting, or replacing nucleotides, or modulating gene expression. Another option is to introduce pathogenic mutations into iPSC lines from healthy individuals to replicate human disease and generate isogenic pairs of cell lines that identify the true impact of the engineered cellular phenotype changes [28].

Finally, it should be noted that iPSC-derived neurons express embryonic features that directly show the genetic mutation effect on the cell phenotype but erase epigenetic changes due to age or disease progress [19]. Although this feature may be useful to identify the specific genetic mutations, it may also hinder the understanding of changes related to time or disease progression. To overcome this potential limitation, in recent years several methods have been implemented for directly reprogramming one somatic cell type into neurons (induced Neurons [iN]) without the transition through iPSCs (transdifferentiation). Transdifferentiated cells have the ability to be identical to their native counterparts; however, in vitro and in vivo assays are necessary to fully characterize them and compare them to native cells [29]. The first method of direct cell reprogramming consists of cell exogenous transgene introduction to overexpress key transcription factors that start the transdifferentiation process [30]. Other methods target endogenous genes that are vital to the transdifferentiation by directly manipulating the DNA or the epigenetic environment [31]. Finally, cells can be treated with pharmacological agents that can modify the genetic and epigenetic environment to promote transdifferentiation [32].

### 2.2. Three-Dimensional (3D) Brain iPSC Culture Models (Organoids)

Brain organoid (BO) is a stem cell-derived 3D tissue that recapitulates early cerebral developmental events and simulates the human brain’s architecture and functionality in vitro. The possibility of generating organoids relies on stem cells’ intrinsic self-organization properties into 3D architectures that contain multiple cell types and maintain some specific function and spatial human brain organization [33,34].

The 3D context, more than 2D culture systems, allows the study of alterations in neuronal migration, cortical layering, and axon guidance that characterize neurodevelopmental and neuropsychiatric disorders, which often do not display observable neuroanatomical phenotypes [35]. Technical advances in 3D culture systems have reproduced different human brain regions [36], and the combination of these independent regional BOs explores complex neuropathological conditions.

The generation of BO began with the development from iPSCs or ESCs of embryoid bodies (Ebs) containing three layers, namely, endoderm, mesoderm, and ectoderm. Neural tissue in humans develops from the ectoderm; thus, Ebs were placed in neural induction media that promote neuronal differentiation after 1 month of culture and then embedded in scaffold support to grow complex organoids. The embedded organoids were transferred into a spinning bioreactor for further maturation and preservation through enhanced nutrient absorption. Over the next 1–2 months, the cerebral tissue gradually expanded to form different brain regions surrounded by cortical tissue [37].

## 3. The Potential of Induced Pluripotent Stem Cells in Personalized Psychiatry

The best medical application of iPSCs is generally thought to be in cell transplantation therapy; however, disease modeling and drug screening could be as relevant as cell therapy, making iPSCs attractive candidates for future PP development. Modeling of iPSC-disease reproduces a pathologic condition in vitro by reprogramming the patient’s somatic cells into iPSCs, followed by redifferentiating the patient-specific iPSCs into disease-specific cells. Additionally, drug screening procedures can use the derived cells from humans for which the test compounds are therapeutically intended.

Meanwhile, iPSCs may be derived from individuals with various monogenic and polygenic disorders thus providing a precious resource for understanding the underlying molecular and cellular mechanisms through personalized models of psychiatric diseases. Moreover, cells are reprogrammed to a very early stage of development; thus, they can provide information on developmental or differentiation defects, as well as the temporal sequence of events, in the early stages of disease progression. The timeline for iPSC differentiation into CNS cells follows the same trajectory as in the developing embryo; thus, these cells are handy tools to study psychiatric disorders, including schizophrenia (SZ) and neurodevelopmental disorders, such as autism spectrum disorder (ASD), where neurodevelopmental alterations are believed to feature. Moreover, BO—which contains multiple organ-specific cell types with function and spatial organization similar to a human brain—has opened up a new avenue for investigating psychiatric disorders.

Finally, iPSCs could offer the possibility of overcoming the ethical concern of destroying a potential human life and the technical limitations related to ESC use. Producing a large number of ESCs or deriving them from patients with diseases is impossible; thus, their use is limited to studies of normal cellular function or the introduction of known engineered genetic changes [32].

In the following paragraphs, we provide an overview of preliminary studies that used iPSCs in severe psychiatric disorders.

### 3.1. Schizophrenia

Schizophrenia (SZ) is a severe mental illness that is associated with subtle brain cortical structure changes and is characterized by the following symptoms: delusions, hallucinations, disorganized thinking, abnormal motor behavior, and negative symptoms [5].

In patients with SZ, iPSC studies are mainly devoted to investigating the pathogenesis of the illness (Table 1).

The first human study by Brennand et al. found diminished neuronal connectivity, neurite number, and PSD95 synaptic protein levels, as well as altered gene expression profiles of the cAMP, wingless-related integration site (Wnt) signaling pathways, and glutamate receptors in iPSC-derived neurons of patients with SZ, compared to controls. The Wnts are secreted factors that regulate cell proliferation and differentiation during embryonic development and act by activating diverse signaling cascades inside the target cells. These alterations are consistent with those described in the postmortem brain of patients with SZ and animal models of SZ and could be ameliorated with the antipsychotic loxapine [38]. More recent studies confirm the presence of early neurodevelopmental alterations. Migration capacity was impaired in induced neural stem cells (iNSCs) from patients with SZ compared to iNSCs from healthy controls and genetic high risk (GHR) individuals [39]. Moreover, iNSCs from a GHR individual who later developed SZ showed migratory impairment similar to SZ-iNSCs.

Narla et al. identify the nFGFR1 signaling, which integrates signals from diverse pathways that are characterized by SZ-linked mutations, as a common altered mechanism and a potential therapeutic target in investigated patients [40].

Hippocampal neurogenesis aberrations have been implicated in SZ pathogenesis [51]. Thus, iPSC-based case/control studies investigated the early development in iPSC-derived hippocampal neurons of patients with SZ. The dentate gyrus (DG) of the hippocampus is one of the two areas of the brain where neurogenesis continues throughout life and its generated neurons play a key role in learning and memory. Yu [41] described reduced neuronal activity and reduced spontaneous neurotransmitter levels released in the dentate granule neurons from SZ-iPSC-derived hippocampal NPCs. Moreover, hippocampal CA3 neurons that are derived from these patients had altered network connectivity when co-cultured with human dentate granule neurons [42].

The mitochondrial tricarboxylic acid (TCA) cycle (also referred to as the Krebs cycle) is known to largely satisfy the energetic neuron demand, thereby generating approximately 90% of cellular reactive oxygen species (ROS). Mitochondrial dysfunction enhances ROS formation, which negatively acts on mitochondrial function, leading to oxidative damage that affects several cellular components, such as lipids, DNA, and proteins. The human fetal brain is particularly vulnerable to oxidative damage due to its high oxygen consumption, high ROS basal level production, and less developed antioxidant defense mechanisms relative to the adult brain. Collectively, the studies in iPSC-NPCs and neurons of patients with SZ showed perturbed mitochondrial respiration and morphology along with signs of increased oxidative damage compared to controls [52].

Globally, SZ iPSC-derived NPCs demonstrated altered migration and Wnt signaling [47], increased oxidative stress [43], and perturbed environmental stressor responses [44].

BO studies in patients with SZ have revealed several phenotypes that may be associated with early neurodevelopmental defects, such as nFGFR1 signaling [45] and immune response alterations [46]. The BOs produced with iPSCs of eight patients with SZ and eight healthy controls were compared with a transcriptomic approach to identify disease-specific differences [47]. Significantly, RNA sequencing demonstrated aberrant gene expression in pathways involved in synaptic biology, nervous system development, immune response, mitochondrial function, and excitatory and inhibitory neurotransmission modulation.

Notaras et al. observed that in SZ BO, about 2.62% of global proteome was differentially regulated, with a depletion of factors that support neuronal development, differentiation, and/or function [48]. Moreover, the differential regulation of two novel specific disease candidates identified through genome-wide association study (GWAS) arrays (namely, Pleiotrophin and Podocalyxin) was observed [48]. The SZ organoids exhibited altered progenitor survival and disrupted neurogenesis, yielding fewer neurons within developing cortical areas [49]. Single-cell sequencing revealed that SZ progenitors were specifically depleted of neuronal programming factors. This study suggests that multiple mechanisms in SZ-derived BO converge upon brain developmental pathways and contribute to raising the risk of developing SZ [49].

Finally, Page [50] identified electrophysiological measures in SZ-iPSC-derived neurons that associate with diagnosis and/or predict the severity of clinical and cognitive features of the adult SZ donor.

In addition, CRISPR-mediated gene editing has contributed to our understanding of risk variants for SZ, which include common single-nucleotide variants (SNVs) with small effect sizes and copy-number variations (CNVs) with greater penetrance. By integrating GWAS and post-mortem brain expression quantitative trait loci (eQTL) studies, common variants that affect SZ risk through regulation of gene expression have been identified. Schrode [53] applied CRISPR-mediated gene editing, activation, and repression technologies to study one putative SZ-eQTL (*FURIN*rs4702) and four top-ranked SZ-eQTL genes (*FURIN*, *SNAP91*, *TSNARE1*, *CLCN3*), demonstrating a synergistic effect between SZ-eQTL genes that converge on synaptic function. The single and combinatorial CRISPRa/i manipulations of common variant loci and genes in human neurons suggest that synergy between risk variants may impact SZ risk.

It should be noted that some of the biological alterations that emerged from iPSC-based studies appeared not to be specific to SZ but to be shared by several disorders. Particularly, studies have identified in disrupted in schizophrenia 1 (DISC1) a gene associated with diverse mental disorders, following the finding that its coding sequence is interrupted by a balanced translocation in a Scottish family, in which the translocation co-segregates with SZ, bipolar disorder, and major depressive disorder [54]. The variety of disorders in subjects that harbor the translocation supports the hypothesis that the translocation leads to a subtle disruption in neural development that predisposes to mental disorders by increasing vulnerability to other environmental and genetic risk factors. An isogenic iPSC model demonstrated that DISC1 gene disruption at the site of the balanced translocation causes loss of expression of longer DISC1 transcripts, which increases baseline Wnt signaling and alters the transcriptional profile of neural progenitor cells and neurons, resulting in neurodevelopmental disorders [54].

Generated cerebral organoids by DISC1-disrupted iPSCs showed disorganized structural morphology and impaired proliferation, which is phenocopied by Wnt agonism and rescued by Wnt antagonism [55]. The shared changes in BO morphology and gene expression with DISC1 interruption and Wnt agonism highlight the link between DISC1 mutation, Wnt signaling abnormalities, and neuropsychiatric diseases.

Overall, the iPSC-based studies in patients with SZ suggest the presence of an early genetic dysregulation leading to altered neuronal and brain development.

### 3.2. Major Depressive Disorder

Major Depressive Disorder (MDD) is a common and severe mental disorder that is characterized by one or more major depressive episodes, defined as discrete periods of at least 2-week duration, but generally longer, in which at least five of the following nine symptoms are present: depressed mood, loss of pleasure or interest, significant appetite disturbance/body weight change, sleep disturbance, loss of energy, psychomotor changes, excessive guilt and/or worthlessness, decreased concentration, and recurring death and/or suicide thoughts [5]. Unsatisfactory response rates to currently approved antidepressant drugs, which are effective in approximately half of the treated patients, contribute to the heavy medical and economic burden of MDD [56].

Therefore, iPSC-based studies on MDD have focused on treatment-resistant depression (TRD), which is defined as the failure to achieve a reduced baseline depressive symptomatology of at least 50%, after at least two antidepressant treatment trials of adequate dosage and duration (Table 2).

A study on patients with MDD showed that iPSCs derived from patients who are non-remitted with selective serotonin reuptake inhibitor (SSRI) antidepressant therapy displayed serotonin-induced hyperactivity downstream of upregulated excitatory serotonergic receptors (5-HT2A and 5-HT7) in contrast to what was seen in healthy and remitted patient-derived iPSCs. Lurasidone, which is a high-affinity 5-HT2A and 5-HT7 antagonist, partially rescued 5-HT-induced hyperactivity in non-remitted patient-derived iPSCs [57].

Another study of the same research group revealed no significant differences in serotonin release/reuptake or genes related to serotonin pathways in iPSCs and serotonergic neurons of SSRI-treated patients with MDD [58]. However, non-remitted patient-derived serotonergic neurons exhibited altered neurite growth and morphology downstream of lowered key *protocadherin-α* gene expression compared to healthy controls and remitted patients [58]. The protocadherin-α family includes cell adhesion molecules that are required for serotonergic projections to appropriately innervate target brain areas, so that their loss from serotonergic neurons leads to unbalanced distributions of serotonergic axons [62].

In MDD lymphoblastoid cell lines (LCLs) reprogrammed to iPSC and differentiated into cortical neurons, Avior [59] identified bupropion response-specific biomarkers, including synaptic connectivity and morphology changes as well as specific gene expression. The LCLs utilized in this platform can be easily obtained from patients, promoting personalized treatment.

Ketamine, a glutamate N-methyl-d-aspartate (NMDA) receptor antagonist, is a promising treatment for patients with TRD, providing significant depressive symptom improvement within hours of intravenous administration. An iPSC-based study of ketamine’s antidepressant mechanisms of action supported the leading hypothesis that ketamine can enhance structural plasticity through the AMPA glutamate receptor-driven raise of Brain-Derived Neurotrophic Factor levels, leading to an increased synaptic number and function in the prefrontal cortex [60].

The mechanisms underlying the prolonged antidepressant effects (1 week) after a single ketamine infusion are poorly understood. The triggering of synaptic function and plasticity was hypothesized because a ketamine half-life of approximately 2 h cannot explain the long-lasting antidepressant effects. A study that supported this view demonstrated that dopaminergic neurons that are differentiated from iPSCs obtained from healthy donors who are exposed for 6 h to ketamine metabolite (2R, 6R)-hydroxynorketamine (half-life 6–12 h) produced dendrite outgrowth when measured 3 days after exposure [61].

### 3.3. Bipolar Disorders

Bipolar disorder (BD) is characterized by the recurring swing between the opposite mood states of major depressive episodes and mania (BD type I) or hypomania (BD type II). Mania is more severe than hypomania and causes markedly impaired social or occupational functioning or necessitates hospitalization. Both manic and hypomanic episodes are characterized by expansive or irritable mood, increased activity and self-confidence, talkativeness, and distractibility, as well as decreased need for sleep, racing thoughts, and poor judgment (5). First-line BD treatment includes mood stabilizers, mainly lithium. The IPSC-based studies on BD are focused on elucidating the pathogenetic mechanism by also examining the detailed effects of lithium and/or other BD drug exposure on the different signaling pathways (Table 3).

The first study that investigated iPSC-derived neuronal cells from patients with BD revealed that the gene expression encoding membrane-bound receptors and ion channels, particularly transcripts involved in calcium signaling, was significantly increased in the neurons generated from three patients with BD compared to those obtained from three healthy controls [63]. Additionally, in vitro lithium pretreatment significantly altered the calcium signaling and electrophysiological properties in BD neurons but not in controls [63]. Notably, calcium signaling has a central role in controlling inappropriate neuronal responses and tonic excitability. A proteomic study in iPSC-derived dorsal anterior forebrain cortical neurons suggested that the molecular lithium-response pathway in patients with BD may function via collapsin response mediator protein-2, which acts to modify neuronal dendrite and dendritic spinal formation [64].

The expression of multiple BD-linked genes that are involved in neuronal development, differentiation, and neuroplasticity are regulated by microRNAs (miRNAs), which are small, non-coding RNAs. In particular, miR-34a is predicted to target genes implicated as risk factors for BD and is reduced by lithium and valproic acid, two mood stabilizers that are widely prescribed to prevent depressive and manic recurrences [73].

An increased miR-34a expression has been detected in postmortem brain tissue, directly induced neuronal cells, and iPSC-derived neuronal cells from patients with BD compared to healthy controls [65]. The overexpression of miR-34a in vitro was reported to decrease CACNB3 and ANK3 gene expression, previously identified as BD risk genes, leading to impaired neuronal differentiation, synaptic protein expression, and neuronal morphology [65]. Afterward, several studies [66,74] investigated the specific risk gene variants in iPSC-derived neuronal cells from patients with BD.

In addition, CXCR4 (CXC chemokine receptor-4) expressing NPCs were analyzed from two BD-affected brothers and their two unaffected parents in a family-based paradigm that has the advantage of controlling for the genetic background [67]. The study revealed that patients with BD, compared with their unaffected parents, displayed multiple phenotypic differences at the neurogenesis level and gene expression critical for neuroplasticity, including Wnt pathway components and ion channel subunits.

Kim et al. studied iPSC-derived neurons obtained from patients with BD type I and their unaffected siblings from an Old Order Amish pedigree with a high BD incidence [68]. The observed significant disease-associated differences in gene expression suggested that RNA biosynthesis and metabolism, protein trafficking, and receptor signaling pathway alterations may play a role in BD pathophysiology.

Accumulating evidence suggests a relationship between imbalanced inflammatory response and BD. Astrocytes participate in the brain inflammatory cascade after being activated by pro-inflammatory cytokines. Vadodaria [69] demonstrated that BD astrocytes are transcriptionally different from controls and induced a reduction in neuronal activity when co-cultured with neurons, even without stimulation. Inflammatory stimulation produced an increased secretion of IL-6 from astrocytes.

Studies on iPSC-derived neuronal cells’ electrophysiological activity may help predict the lithium response and develop novel drugs for BD treatment. Differential hyperexcitability responses to in vitro lithium treatment in iPSC-derived hippocampal dentate gyrus-like neuronal cells from six patients with manic BD type I and four unaffected individuals were detected. Particularly, the hyperexcitability phenotype in BD was selectively reversed by lithium treatment only in neurons that are derived from patients who responded to lithium treatment, suggesting that this model of iPSCs might help develop new drugs [70]. Notably, the electrophysiological data obtained from iPSC-derived neuronal cells treated with lithium was used to train an algorithm that can predict the lithium responsiveness of a new patient with a success rate of over 92% [70].

Osere [71] explored the molecular effects of the three most used mood stabilizers (lithium, valproic acid, and lamotrigine) in iPSC-derived NPCs from healthy controls, Li responders, and Li non-treated BD patients. The study suggested that three mood stabilizers with different mechanisms of action affect a specific set of genes, possibly featuring high relevance for the drug mood stabilizing properties.

The first comprehensive study that compared BO generated from patients with BD type I and healthy individuals demonstrated transcriptomic differences with gene downregulation involved in cell adhesion, neurodevelopment, and synaptic biology, along with the gene upregulation involved in immune signaling in patients with BD type I [72].

### 3.4. Autism Spectrum Disorder

Autism Spectrum Disorder (ASD) refers to a broad range of lifelong neurodevelopmental conditions characterized by impaired social abilities and cognitive functions. ASD can be of unknown polygenic etiology (idiopathic) or a specific syndromic disorder, such as Fragile X, caused by a single gene mutation [75].

Since iPSCs recapitulate aspects of the neuronal development process while preserving the patient’s genetic background, they are frequently used to model idiopathic ASD, for clarifying pathogenetic mechanisms (Table 4).

The increased proliferation and differentiation abnormalities of iPSC-derived NPCs from patients with idiopathic ASD with macrencephaly compared to non-ASD controls with normal brain size were suggested to result in early brain overgrowth [76]. The treatment with the insulin-like growth factor 1(IGF-1), a compound currently in clinical trials, partially rescued the observed neuronal network abnormalities [76]. Schafer et al. followed iPSCs from patients with idiopathic ASD with early brain overgrowth during their neuronal development to examine when and how the earliest ASD-specific abnormalities arise [77]. Changes associated with ASD, involving temporal dysregulation of specific gene networks and morphological growth acceleration, tracked back even before the neuronal stage in neural stem cells (NSCs). Bypassing NSC-like stages by direct ASD iPSC conversion into neurons prevents the manifestation of the observed neuronal ASD-associated phenotypes phenotypes [77].

Using gene expression analyses on patients with ASD iPSC-derived BO, Mariani et al. showed the overexpression of the gene FOXG1, which generates an overproduction of GABAergic neurons, and, in turn, an increased brain volume and imbalanced excitation and inhibition systems in the developing cortex [78]. Unfortunately, studies on idiopathic ASD NPCs show conflicting results relating to the proportion of GABAergic inhibitory precursors compared to glutamatergic precursors [83]. Recently, in idiopathic ASD without macrocephaly, impaired neural differentiation was demonstrated in the absence of altered proliferation, in contrast to previous studies [79].

Fragile X syndrome (FXS) is the most prevalent single-gene form of ASDs and is characterized by cognitive impairment, defective communication, hyperactivity, and impulsivity. In particular, FXS is caused by transcriptional FMR1 gene silencing on the X-chromosome during embryonic development with the consequent loss of Fragile X Mental Retardation Protein (FMRP) expression. This FMRP is a selective RNA-binding protein that regulates the translation of many genes at the synaptic sites. The lack of FMRP leads to aberrant differentiation in human iPSC-derived neural progenitor cells [80].

Conventional bidimensional and 3D FXS models based on isogenic FMR1 knock-out mutant iPSCs display altered cortical neuron gene expression and impaired differentiation compared with the wild-type human iPSCs. Cortical BO models show an increased number of glial cells, such as astrocytes, and bigger organoid size, which suggests that FMRP is required to correctly support neuronal and glial cell proliferation and the correct excitation/inhibition ratio in human brain development [81].

Nonsense-mediated RNA decay (NMD) is a cellular surveillance pathway that safeguards the quality and stability of mRNA transcripts by targeting them for degradation if altered. Deficiency in FMRP results in hyperactivated NMD in FXS fibroblast-derived iPSCs, with a negative consequence on iPSC maturation to neurons [82].

### 3.5. Conclusions

The iPSC technology allows the capturing of the SZ genotype, including genetic risk factors, along with their effects on cellular and molecular endophenotypes during early neurodevelopment. This approach supports the neurodevelopmental hypothesis of SZ, stating that the disruption of early brain development increases the risk of later manifest psychotic symptoms. Patient-derived iPSCs have proven to be a powerful tool in identifying SZ early neurodevelopmental defects, having revealed alterations in neuronal differentiation [45,48,49], migration capacity [39,43], neurite number and length [38,66], synaptic biology [47,49,50], connectivity [38], and neuronal activity [41,42,50].

Additionally, the studies showed perturbed mitochondrial respiration function and morphology associated with signs of increased oxidative damage in SCZ iPSC-NPCs and neurons relative to controls [5,47]. These results agree with neuroimaging, postmortem brains, and patient-derived cells, which have implicated mitochondrial dysfunction in SZ pathogenesis. It is generally accepted that prenatal exposure of the developing brain to various environmental challenges increases susceptibility to later SZ, by interacting with a genetic predisposition. As the mechanisms underlying this process remain obscure, iPSC studies evaluated the role of possible factors such as the heat shock factor 1 [49] and the tumor necrosis factor [49]. Further research should be devoted to finding biomarkers indicative of the early SZ stages to allow the delivery of treatment acting on the damaged neurons before neural impairment cannot be reversed.

In the development of MDD, environmental factors play a central role, to the extent even of inducing epigenetic modifications, which are lost during the reprogramming process. This loss may hinder identifying the multiple pathways involved in MDD pathogenesis and may impact the study of antidepressant resistance, one of the main focuses of the iPSC approach to MDD. It is known that environmental factors, particularly childhood maltreatment, are associated with poor treatment response in MDD. Despite these limitations, iPSC studies are providing important information on the factors involved in SSRI resistance [57] and in response to bupropion and ketamine [59,60,61]. It is essential to consider that iPSC experimentations have used developmentally-immature neurons; thus, it is unclear whether the observed iPSC-derived phenotypes persist into adulthood in humans.

Turning to BD, this is a complex illness with heterogeneous clinical presentation. To better understand its biological underpinnings, the iPSC approach focused on clinical phenotypes, such as lithium-responsive patient samples, or on populations with similar genetic profiles, such as BD individuals within a family or larger pedigree. As imaging and postmortem studies suggest a neurodevelopmental etiology of BD, with neuroanatomical abnormalities often present at the first episode, some iPSCs investigations addressed this. In this context, BD-derived iPSC studies support the presence of early pathologic changes, suggested by the differences between the BP and control iPSCs, in genes involved with proliferation, regulation, and differentiation processes [63,65]. Moreover, BD iPSCs had a lower propensity for generating central neural progenitors, which, in turn, had lower proliferation rates [67]. Additionally, several data point toward aberrant calcium signaling [63,67] and electrical properties [70] in BD neurons, some of which were reduced by lithium although only in the neurons of lithium-responsive patients [70,72]. As lithium is the first-line treatment of BD, reliably predicting lithium response using molecular markers would allow earlier initiation of effective therapy, an approach with demonstrated positive effects on the outcome.

It has been hypothesized that idiopathic macrocephalic ASD might be generated from early proliferation abnormalities that can lead to long-lasting differentiation defects. Increased NPC proliferation leading to functional defects in neuronal networks [76] and dysregulation of GABAergic/glutamatergic pathway differentiation [78] were detected in idiopathic macrocephalic ASD patient iPSC-derived neurons. Of note, treatment with IGF-1, a compound currently in clinical trials, partially rescued the observed neuronal network abnormalities, opening potential therapeutic opportunities for ASD [76]. Changes associated with ASD tracked back to even before the neuronal stage in neural stem cells [77]. In addition, iPSCs from ASD without macrocephaly displayed impaired neural differentiation in the absence of altered proliferation, in contrast to previous studies [79]. Until now, studies on iPSCs derived from patients with idiopathic ASD have been limited by small sample sizes (no more than ten individuals carrying an ASD diagnosis) and varied timing as well as methods of cellular profiling of phenotype. Moreover, ASD is phenotypically and etiologically heterogeneous, making detecting the condition’s underlying pathophysiology arduous.

## 4. Limitations of iPSC Technology and Future Perspectives

Although iPSC models provide an innovative tool to understand the pathophysiology of neuropsychiatric disorders and perform drug screening in disease-relevant cells, several limitations remain. Studies based on iPSC technology are impacted by variation in reprogramming and neuronal differentiation efficiencies between iPSC lines derived from both the same and different donors. Two main strategies have been developed to face this variation. The first tries to decrease the heterogeneity by selecting patients with shared clinical and biological characteristics or drug responses, with the expectation of inter-individual variation reduction in vitro. The second strategy uses a large cohort of disease-relevant iPSC cell lines to minimize sample variability. This effort is costly and time-consuming, thus iPSC banks were created to provide repositories from which many iPSC lines that are available for a particular disease are stringently checked for quality [84].

One critical issue with iPSC-based disease modeling is the generation of appropriate control iPSCs. Initially, control iPSCs were obtained from healthy, gender-matched family members, but these iPSCs exhibited substantial heterogeneity due to genetic differences. More recently, genetically identical (isogenic) iPSC lines that are created by gene editing approaches from well-characterized healthy control iPSC lines have been employed.

A further drawback of iPSC technology refers to the reprogramming process itself, which erases the epigenetic memory of cells. In particular, iPSC-derived neurons are immature and maintain the fetal neuron properties, independently of the age of the initial somatic cell donor [85]. This aspect represents a limit for both the adult-onset psychiatric disorders and those which are influenced by environmental factors that are known to modify epigenetics. Therefore, iPSC-derived models are an opportunity for studying susceptibility rather than normal disease or disease progression [35]. Techniques to artificially induce age in iPSC-derived lineages for modeling late-onset disorders, such as the exposure to compounds that trigger mitochondrial stress or reactive oxygen species, have been developed [85].

The discovery of direct reprogramming technology has enabled induced neuron (iNs) production, bypassing the iPSC stage. This approach produces a reliable model for late-onset psychiatric disorders, aging-related neurodegenerative diseases, and drug discovery as it does not reset aging information [86]. Unfortunately, iN, unlike iPSC, does not maintain self-renewal, which is required for maintenance and stock; hence, the technology relies on acquiring a larger quantity of original cells from a patient to obtain enough iNs.

A central limitation of BO is the absent development past the stage of a prenatal brain, probably because the organoids do not elaborate a vascular system. This inadequacy additionally leads to necrotic core formations during tissue development. Therefore, engineering approaches that allow the proper exchange of oxygen, nutrients, and waste products, such as porous scaffolds, recently developed using computer-aided design and 3D printing, have been implemented to address this issue [87].

Another constraint, especially for 3D cultures, is the extensive period involved. Achieving later stages of cell maturation, including astrocyte maturation, takes approximately 9 months. This challenges the feasibility of using these 3D cultures on a very large scale and considerably slows the experimental processes [88].

From a therapeutic perspective, gene-editing techniques could be used to repair the genetic mutations that contribute to disease in an in vivo approach, which involves direct cell modification in the individual, and to repair known causative lesions in patient-derived iPSCs (ex vivo gene therapy). This last intervention requires the culture and modification of patient-derived iPSCs in vitro and the transplantation of the modified cells back into the recipient [27].

For neurodegenerative disorders, such as Parkinson’s disease, iPSC replacement therapies are challenged by the transplantation into an environment where pathogenic mechanisms occur, thereby causing neuronal degeneration and apoptosis. In theory, inserting healthy cells in this type of environment can either have a positive effect due to the neuroprotective factor secretion or result in graft failure. In a recent study, a personalized cell-therapy strategy that used iPSC-derived dopaminergic cells in a patient with Parkinson’s disease lead to a clinical improvement 24 months post-transplantation, a time frame consistent with gradual putamen reinnervation by projections from dopaminergic neurons [89].

A more recent and innovative advancement in iPSC technology is the development of the organ-on-a-chip (OOC) platforms that introduce microfluidic techniques in cell cultures to reproduce the microenvironment through the control of biochemical factors, mechanical forces, and fluid flow. In particular, OOC aims to reflect individual pathophysiological conditions by including blood samples and patient-derived iPSCs and by adjusting biochemical parameters according to personal health data [90]. This strategy could open new ways to build a disease model that considers individual variability and should lead to a “personalized” representation of the individual patient and therefore be directly applied in the clinic to inform targeted prevention and treatment strategies. However, personalized OOC has not yet been applied in clinical practice. The key steps for future successful implementation include the demonstration that OOC has added value in directing personalized treatment and prevention strategies.

It is essential to highlight that even the most innovative PP advances should be understood not as a replacement but as an integration of patient assessment. Every therapeutic choice must be inserted into the individual patient’s reality in light of his/her personal and clinical history, complying, to the extent possible, with their own preferences regarding the type of intervention.

## Figures and Tables

**Table 1 jpm-12-01340-t001:** Selected studies investigating schizophrenia using iPSCs from patients.

In Vitro Model	Observation (s)	Implication (s)	References
The iPSCs differentiated to NPCs, glutamatergic neurons	Diminished PSD95 synaptic protein levels, altered expression of Wnt signaling pathway genes	Diminished neuronal connectivity, neurite number, altered Wnt signaling pathways, and glutamate receptors	Brennand et al., 2011 [38]
Genetic high risk (GHR) individuals and SZ-iNSC-derived without genetic modification	The migration rate of SZ-iNSCs was significantly slower than that of HC-iNSCs	Migration capacity was impaired in SZ-iNSCs compared to iNSCs from GHR individuals or controls. iNSCs from a GHR individual who later developed SZ showed migratory impairment similar to SZ-iNSCs.	Lee et al., 2022 [39]
The iPSCs differentiated to neuron committed cells (NCCs)	Altered nFGFR1 signaling which integrates diverse pathways with schizophrenia-linked mutations	Global dysregulation of developmental genome	Narla et al., 2017 [40]
The iPSC-derived DG hippocampal neuron	Deficits in the generation of DG granule neurons	Reduced neuronal activity and spontaneous neurotransmitter release	Yu et al., 2014 [41]
The iPSCs, differentiated to CA3 neurons and DG neurons	Reduced activity in DG-CA3 co-culture and deficits in spontaneous and evoked activity in CA3 neurons	iPSC-derived DG-CA3 co-cultures present deficits in hippocampal activity	Sarkar et al., 2018 [42]
The iPSC-derived neurons neural progenitor cells (NPCs).	Abnormal gene expression and protein levels related to cytoskeletal remodeling and oxidative stress	Aberrant migration and increased oxidative stress	Brennand et al., 2015 [43]
The NPCs derived from SZ-iPSCs and HC	The embryo exposure to environmental factors activates HSF1 in cerebral cortical cells. SZ-NPCs showed higher variability in the levels of HSF1 activation than HC	HSF1 plays a role in the response of brain cells to prenatal environmental insults	Hashimoto-Torii et al., 2015 [44]
Embryonic stem cell (hESC) and iPSC-derived BO	The nFGFR1, that controls the brain development by integrating signals from diverse development–initiating factors, was lost from the nuclei of differentiating cortical neurons in SZ-BO	The loss of nFGFR1 likely underlies the SZ impaired cortical neuronal differentiation	Stachowiak et al., 2017 [45]
Embryonic stem cells (ESC), SZ- and HC-iPSC-derived BO	In HC-BO iPSC the transient exposition to TNF produced malformations similar to those of the SZ-BO. Both SZ- and TNF-induced malformations were associated with the loss of nFGFR1 in the cortical zone	Maternal infection during pregnancy exposes the fetus to elevated TNF levels. The TNF, similar to SZ, alters nFGFR1 signaling, NPCs, and neurons in developing BO	Benson et al., 2020 [46]
An SZ-iPSC and HC-iPSC derived BO	Downregulation of genes involved in cell adhesion, neurodevelopment, and synaptic biology along with the upregulation of genes involved in immune signaling in SZ-BO compared to HC-BO	The SZ-BO showed differences in expression of genes involved in mitochondrial function, excitation/inhibition balance modulation, synapse biology, neurodevelopment, and immune response	Kathuria et al., 2020 [47]
An SZ-iPSC and HC-iPSC derived BO	About 2.62% of global proteome was differentially regulated in SZ organoids (43 proteins up-regulated and 54 down-regulated) with depletion of factors that support neuronal development and differentiation	Key pathways regulating nervous system development was perturbed in SZ-derived organoids	Notaras et al., 2021 [48]
An SZ-iPSC and HC-iPSC derived BO	The SZ organoids exhibited progenitors depleted of neuronal programming factors leading to altered developing cortical areas	Multiple mechanisms in SZ-derived BO converge upon brain developmental pathways and contribute to raising the risk of developing SZ	Notaras et al., 2022 [49]
The iPSC-derived neural progenitors and cortical neurons	Electrophysiological measures in iPSC-derived neurons showed altered Na+ channel function, action potential interspike interval, and GABAergic neurotransmission	Electrophysiological measures predicted cardinal clinical and cognitive features	Page et al., 2022 [50]

SZ: schizophrenia; HC: healthy controls; BO: brain organoid; DG: dentate gyrus; iNSCs: induced neural stem cells; NPCs: neural progenitor cells; Wnt: wingless-related integration site signaling pathways; GABA: gamma-aminobutyric acid; nFGFR1: nuclear FGFR1; HSF1: heat shock factor 1; CNVs: copy number variations; TNF: tumor necrosis factor.

**Table 2 jpm-12-01340-t002:** Selected studies investigating major depressive disorder using iPSCs from patients.

iPSC Model (s)	Observation (s)	Implication (s)	References
The iPSCs from SSRI-remitters and differentiated to serotonergic neurons	The SSRI-non-remitters patient-derived neurons displayed 5-HT-induced hyperactivity via upregulated 5-HT2A and 5-HT7 receptors	The SSRI-resistant patients represent a subset of MDD patients with differential responses to 5-HT	Vadodaria et al., 2019 [57]
The iPSCs from SSRI-remitters and SSRI-non-remitters differentiated to serotonergic neurons	Altered morphological phenotypes associated with SSRI-non-remission downstream of PCDHA6 and PCDHA8 genes	The differences in serotonergic neuron morphology may contribute to SSRI-non-remission in MDD patients	Vadodaria et al. [58]
The MDD BP-responders’ and- non-responders’ LCLs, reprogrammed to iPSCs, and differentiated to mature prefrontal cortex neurons	Following BP treatment, cortical neurons of BP-responders, compared with vehicle treatment, had significant differences, including enhanced colocalization of synaptic markers and spine morphology	The enhanced colocalization of synaptic markers observed in BP-responders’ derived cells could be indicative of enhanced connectivity following effective antidepressant treatment. The synaptic changes could be used as a biomarker for BP effects in vitro	Avior et al., 2021 [59]
The iPSC-derived dopaminergic neurons	Ketamine elicits structural plasticity in iPSCs-derived DA neurons by recruitment of AMPA, mTOR and BDNF signaling	The transient exposure to ketamine dose-dependently promotes structural plasticity as determined by enhanced dendritic outgrowth and increased soma size	Cavalleri et al., 2018 [60]
The iPSCs from healthy donors differentiated to dopaminergic neurons	Exposure to the ketamine metabolite HNK for 6 h produces AMPA receptor-mediated and mTOR-mediated dendrite outgrowth when measured 3 days after exposure	The prolonged antidepressant. Effect after a single infusion of ketamine is based on the triggering of long-lasting neuroplasticity	Collo et al., 2018 [61]

SSRI: selective serotonin reuptake inhibitor; BP: bupropion; LCLs: lymphoblastoid cell lines; AMPA: α-amino-3-hydroxy-5-methyl-4-isoxazolepropionic acid; BDNF: brain-derived neurotrophic factor; mTOR: mammalian target of rapamycin; DA: dopaminergic; HNK: (2R,6R)-hydroxynorketamine.

**Table 3 jpm-12-01340-t003:** Selected studies investigating bipolar disorder using iPSCs from patients.

iPSC Model (s)	Observation (s)	Implication (s)	References
The iPSCs differentiated to neurons	Increased expression of transcripts for membrane-bound receptors and ion channels in BD-derived neurons than in controls	Neurons from BD iPSC were significantly different in their gene expression, particularly transcripts involved in calcium signaling, from those derived from control iPSC	Chen et al., 2014 [63]
The iPSCs differentiated to neurons	Lithium alters the phosphorylation state of collapsin response mediator protein-2 (CRMP2)	The molecular lithium-response pathway in BD acts via CRMP2 to alter neuronal cytoskeletal dynamics, mainly dendrite and dendritic spine formation/function	Tobe et al., 2017 [64]
The iPSC-derived NPCs	The miR-34a is upregulated and targets BD risk genes ANK3, CACNB3, and DDN	The miR-34a overexpression impairs neuronal differentiation, expression of synaptic proteins and neuronal morphology	Bavamian et al., 2015 [65]
The iPSCs from patients who carried CNVs differentiated to neurons	Exonic deletion of Protocadherin 15 (PCDH15) is associated with shorter dendrites and decreased number of synapses than controls in both glutamatergic and GABAergic neurons	Neurons derived from iPSCs exhibited dendrite shortening and decreased synapse numbers	Ishi et al., 2019 [66]
The CXCR4 expressing NPCs	Phenotypic differences at the level of neurogenesis and expression of genes critical for neuroplasticity, including WNT pathway components and ion channels	Abnormalities in early steps in NPC formation, WNT/ GSK3 signaling and expression of ion channels in the BD patient-derived NPCs and neurons	Madison et al., 2015 [67]
The iPSCs differentiated to NPCs and then to neurons from Old Order Amish of Lancaster County	A total of 328 genes were differentially expressed between BPD and control L neurons including GAD1, glutamate decarboxylase 1, and SCN4B, the voltage gated type IV sodium channel beta subunit	The differentially expressed genes suggested that the alterations in RNA metabolic processes, protein trafficking, and receptor-mediated signaling contribute to BD biology	Kim et al., 2015 [68]
The GPCs from BD- and HC-iPSCs	The BD astrocytes induced a reduction in neuronal activity when co-cultured with neurons, mainly after cytokine stimulation	The BD astrocytes are functionally less supportive of neuronal activity and this effect is partially mediated by cytokines	Vadodoria et al., 2021 [69]
The iPSCs differentiated to hippocampal DG-like neurons	Using both patch-clamp recording and somatic Ca^2+^ imaging, hyperactive action-potential firing was observed	The hyperexcitability phenotype of BD neurons was reversed by lithium in neurons derived from patients who also responded to lithium treatment	Mertens et al., 2015 [70]
The iPSC-derived NPCs	Gene expression signature associated with the three most commonly used mood stabilizers (Li, VPA, and LTG)	The findings support the involvement of mitochondrial functions in the molecular mechanisms of mood stabilizers	Osere et al., 2021 [71]
The BD-iPSCs derived from EBV immortalized lymphocytes differentiated to DG granule neurons	Endophenotype of hyperexcitability is shared by BD DG neurons. Functional analysis showed that intrinsic cell parameters are very different between neurons derived from Li-responders and those derived from Li-non-responders	Chronic Li treatment reduced the hyperexcitability in the lymphoblast-derived Li-responders but not in the Li-non-responders	Stern et al., 2018 [72]

NPCs: neural progenitor cells; DG: dentate gyrus; Li: lithium; VPA: valproate; LTG: lamotrigine; CNVs: copy number variations; CXCR4: CXC chemokine receptor-4; GPCs: glial progenitor cells; HC: healthy controls.

**Table 4 jpm-12-01340-t004:** Selected studies investigating autism spectrum disorder using iPSCs from patients.

iPSC Model (s)	Observation (s)	Implication (s)	References
The iPSCs differentiated to NPCs and neurons from ASD individuals with early brain overgrowth and non-ASD controls with normal brain size	The ASD-derived NPCs display faster proliferation than control-derived NPCs due to dysregulation of a β-catenin/BRN2 transcriptional cascade	Abnormal neurogenesis and synaptogenesis leading to functional defects in neuronal networks that could be rescued by the neurotrophic factor IGF-1	Marchetto et al. [76]
The iPSCs from ASD with macrocephaly differentiated to NSCs; iPSCs directly converted into iNs; generation of cerebral organoids	Dysregulation of specific transcriptional networks that caused aberrant neuronal maturation of ASD cortical neurons	The ASD-associated neurodevelopmental aberrations are triggered by a pathological priming of gene regulatory networks during early neural development	Shafer et al., 2019 [77]
Organoids derived from ASD-iPSCs	The ASD-derived organoids exhibit an overproduction of GABAergic inhibitory neurons, due to the overexpression of the transcription factor FOXG1	Cortical organoids of ASD patients show exuberant GABAergic differentiation and no change in glutamate neuron types, which together cause an imbalance in glutamate/GABA neuron ratio	Mariani et al., 2016 [78]
The iPSCs from ASD without macrocephaly differentiated to cortical and midbrain neurons	The ASD-iPSCs differentiated to cortical neurons displayed impaired neural differentiation. These cellular phenotypes occurred in the absence of alterations in cell proliferation during cortical differentiation, in contrast to previous studies	Patients with ASD but without macrocephaly exhibited impairments in neurogenesis compared with those from neurotypical individuals	Adhya et al., 2021 [79]
The NPCs derived from *FMR1*-knockout iPSCs as a model for studying FMRP functions and FXS pathology	Altered expression of neural differentiation markers, MRP-deficient neurons showed less spontaneous calcium bursts, corrected by the protein kinase inhibitor LX7101	Loss of FMRP resulted in abnormal differentiation accompanied by impaired neuronal activity	Sunamura et al. [80]
Both 2D and 3D FXS models based on isogenic FMR1 knock-out mutant and wild-type human iPSC lines	Cortical neurons derived from FMRP-deficient iPSCs exhibit altered gene expression and impaired differentiation when compared with the healthy counterpart	The FMRP is required to correctly support neuronal and glial cell proliferation, and to set the correct excitation/inhibition ratio in the developing brain	Brighi et al. [81]
Human SH-SY5Y neuroblastoma cells and FXS fibroblast-derived iPSCs	The FMRP deficiency results in hyperactivated nonsense-mediated mRNA decay (NMD). The key NMD factor UPF1 binds directly to FMRP, promoting FMRP binding to NMD targets	The FMRP acts as an NMD repressor. In the absence of FMRP, NMD targets are relieved from FMRP-mediated repression. Many abnormalities in FMRP-deficient cells are attributable, either directly or indirectly, to misregulated NMD	Kurosaky et al. [82]

ASD: Autism Spectrum Disorder; FXS: Fragile X syndrome; NSCs: neural stem cells; iNs: induced neurons; IGF-1: insulin-like growth factor 1; FMRP: fragile X mental retardation protein.

## Data Availability

Not applicable.

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
