# Peer review of "Human-Induced Pluripotent Stem Cell Technology: Toward the Future of Personalized Psychiatry"

_jpm, 2022, doi:10.3390/jpm12081340_

Round 1
Reviewer 1 Report
The review manuscript "Human-induced pluripotent stem cell technology: toward the future of personalized psychiatry" by A. Alciati et al. describes application of iPSCs and their derivates in the study of the pathogenesis of such multifactorial diseases as schizophrenia, major depression disorder, bipolar disorder, and autism spectrum disorder, as well as in assessment of drug efficiency. The review is well written, understandable, easy to read, and quite complete.
I am confident that after correcting major or minor changes, this manuscript can be published in the Journal of Personalized Medicine.
Major comments:
1. There are no illustrations in the manuscript. Perhaps the authors consider it possible to add at least 1-2? For example, obtaining iPSCs and their differentiation in the neuronal direction. Or maybe the authors will represent schemes of some molecular pathways discovered using iPSCs, violations in which led to the development of some diseases.
2. Line 460. The authors need to consider the issue of incomplete differentiation of iPSCs into derivatives and discuss how it is possible to obtain a cell line that does not contain iPSCs to minimize the risk of developing teratomas.
Minor comments:
1. Line 18. Please delete “induced pluripotent stem cells”.
2. Line 100. Please specify which virus?
3. Line 121. Authors should mention mRNA-based method for reprogramming somatic cells.
4. Line 128. Cas9 is nuclease, but not endonuclease.
5. Line 128-130 “Cas9 nuclease can target specific DNA sequences with the help of guide RNA, can be used to target and correct any specific gene sequences” – Cas9 with sgRNA cannot correct any specific gene sequences themselves. They definitely need a system for repairing double-stranded DNA breaks in the cell and often a donor molecule for recombination.
6. Can the authors provide the results of studies, if any, on genome editing of CNS cells derived from iPSCs? I understand that the focus of the article is on multifactorial diseases, but perhaps there are scientific studies in this area using, for example, CRISPRa or CRISPRi for activation of inhibition some genes, involved in pathogenesis of psychiatric disorders.
Reviewer 2 Report
Dear Authors,
Induced pluripotent stem cells (iPSCs) technology is a source of knowledge about the mechanisms of complex neuropathological conditions, including mental disorders. iPSCs can be used for disease modeling and drug screening. Your review article “Human-induced pluripotent stem cell technology: toward the future of personalized psychiatry” is interesting and may have impact on this area of research. In this review, you described the prospects for iPSCs technology to model and to develop treatments for schizophrenia, major depressive disorder, bipolar disorder, and autism spectrum disorder. The manuscript is well written and understandable.
However, this review has a number of shortcomings. I have major suggestions for improving the manuscript:
1. The main limitation of this review is that sections 3.1, 3.2, 3.3, and 3.4 simply list some data from studies using iPSCs technology in schizophrenia, MDD, BD, and ASD. No systematization of information and synthesis of knowledge have been made. There are practically no conclusions. In each of the four sections 3.1, 3.2, 3.3, and 3.4, I recommend adding tables to organize the observations. Each table may contain the following sections: Sample origin, Cohort, In vitro model (Reprogramming Technique, iPSC differentiation), Observation(s), Implication(s), References, etc., for example, as in https://doi.org/10.3389/fnmol.2021.756613 (Table 2).
You can also add a comparison table of observed anomalies of iPSCs in schizophrenia, MDD, BD, and ASD, for example, as in https://doi.org/10.1002/sctm.20-0206 (Table 2).
2. Research on iPSCs in psychiatric disorders is advancing rapidly. This review considers works mainly from 2000-2020. New works for 2021-2022 are practically not considered. Even a quick search reveals a number of new papers on iPSCs in psychiatric disorders:
https://doi.org/10.1016/j.tins.2021.11.002
https://www.pnas.org/doi/10.1073/pnas.2109395119
https://www.nature.com/articles/s41398-021-01319-5
https://www.nature.com/articles/s41380-021-01164-4
https://doi.org/10.1016/j.stemcr.2021.02.004
I advise you to conduct a more thorough search for recent literature.
A systematic review that analyzes iPSC models in neuropsychiatric research will also help you to find more research on this topic:
https://link.springer.com/article/10.1007/s00702-020-02197-9
(I do not call for citing the above works. They are provided for your reference. I assure you that I have no interest in citing these works.)
3. You need to add a Conclusion section. It should contain conclusions from the analyzed studies and summarize all the data. You can also identify unresolved issues that need to be investigated.
I have also minor suggestions for improving the manuscript:
Line 28 – “major depression disorder” – I hope this is a typo because this is an incorrect term, the correct one would be “Major depressive disorder”.
Line 105 – “Human iPSCs were first established from skin fibroblasts through a skin biopsy” – Please add a reference.
Line 187 – “schizophrenia (SCF)” – for schizophrenia, the abbreviation SZ or SCZ is most commonly used. SCF is a rather strange abbreviation (there is no “F” in the word). Please correct here and further in the text.
Line 198 – “IPSCs” – Please replace with “iPSCs”.
Line 203 – “IPSCs” – before that, you used the abbreviation “iPSCs” at the beginning of a sentence. Please use the uniform abbreviations.
Line 441 – “iNs” – Please define an abbreviation.
Good luck with your further research.
Best regards
